# Communication-Efficient Vertical Federated Learning

Afsana Khan *, Marijn ten Thij(orcid) and Anna Wilbik(orcid)

Department of Advanced Computing Sciences, Maastricht University, 6229 EN Maastricht, The Netherlands
* Correspondence: a.khan@maastrichtuniversity.nl

**Abstract:** Federated learning (FL) is a privacy-preserving distributed learning approach that allows multiple parties to jointly build machine learning models without disclosing sensitive data. Although FL has solved the problem of collaboration without compromising privacy, it has a significant communication overhead due to the repetitive updating of models during training. Several studies have proposed communication-efficient FL approaches to address this issue, but adequate solutions are still lacking in cases where parties must deal with different data features, also referred to as vertical federated learning (VFL). In this paper, we propose a communication-efficient approach for VFL that compresses the local data of clients, and then aggregates the compressed data from all clients to build an ML model. Since local data are shared in compressed form, the privacy of these data is preserved. Experiments on publicly available benchmark datasets using our proposed method show that the final model obtained by aggregation of compressed data from clients outperforms the performance of the local models of the clients.

**Keywords:** federated machine learning; heterogeneous federated learning; communication efficient; data privacy





## 1. Introduction

As organizations seek to modernize and optimize business processes, machine learning (ML) has emerged as a powerful tool for driving automation. It has aided in the enhancement of business scalability and the improvement of business operations for companies all over the globe by extracting meaningful insights from raw data to quickly solve complex, data-rich business problems. Furthermore, in healthcare, machine learning (ML) applied to electronic health records (EHRs) can yield actionable insights, ranging from improving patient risk score systems to predicting disease onset and streamlining hospital operations [1]. ML has also made a significant contribution to the agriculture sector by assisting farmers in reducing farming losses by providing rich recommendations and insights into crops [2]. There is virtually no application domain that could not benefit from using ML techniques for decision support. Although organizations benefit from using machine learning approaches on their data, using data from other similar organizations for the same purpose could result in significant improvements to the existing organizational processes. Recognizing the importance of collaboration, a significant emphasis has been placed on integrating data from multiple organizations in order to design sophisticated machine learning models for improving customer service and acquisition.

However, at present, data sharing among organizations has become critical due to concerns of privacy, maintaining competitive advantages, and/or other constraints. Although significant research has been conducted related to distributed learning [3], which aims at performing tasks on data distributed across multiple servers, it mainly focuses on reducing the time required to perform tasks by parallelizing computation power. On the other hand, federated learning (FL) focuses on data locality [4] and is seen as a promising technology approach that enables a network of autonomous organizations that face the same machine learning task to collaboratively learn a global model that offers better predictive performance for all participants without the need to share sensitive data. Generally, FL

can be divided into different scenarios based on how the data are partitioned or distributed among the data owners, i.e., horizontally or vertically. Horizontal federated learning (HFL), also known as Homogeneous FL, is used in scenarios where data owners possess data with the same characteristics or features, but differ in the number of samples they possess in their data. An example of HFL is a group of hospitals collaborating to build a model that can predict a health risk for their patients, based on agreed data.

However, this is not the only possibility. Imagine a scenario in which multiple data owners or clients wish to collaborate in the training of ML models having common samples of data, but not the same features, for instance, a telecom company collaborating with a home entertainment company (cable TV provider), or an airline collaborating with a car rental agency. As the data are vertically partitioned rather than horizontally, HFL does not fit in such cases. Vertical federated learning (VFL) can overcome client data heterogeneity (same samples with different features). It can be referred to as the process of aggregating different features and computing the training loss and gradients in a privacy-preserving manner to build a model with data from both parties collaboratively. Although FL comes with a lot of advantages, one of the major challenges of this approach is the communication overhead, which is discussed further in Section 2.

Our key contribution in this paper is the introduction of a communication-efficient vertical federated approach that uses a feature extraction technique to compress local data of clients. The compressed data from the clients are then aggregated to train the final machine learning model. As a result, clients can collaborate by sharing compressed (latent) representations of their raw data without jeopardizing their privacy and security. The whole iteration is just performed once, which reduces the amount of communication needed significantly. Furthermore, our contributions include extensive experiments using the proposed approach on four benchmark datasets and comparing its performance to a centralized machine learning model. The rest of the paper is organized as follows: Section 2 discusses the existing methods designed to reduce the communication overhead in FL environment, Section 3 elaborates on our proposed method in detail, Section 4 explains the experimental setup of the performed experiments, Section 5 demonstrates the results of the experiments and provides a brief analysis. Finally, Section 6 concludes the paper by mentioning the overall observations from the experiments, while also providing insights for possible future improvements.

## 2. Background

### 2.1. Vertical Federated Learning

Vertical federated learning (VFL) is the collaborative training of a model on a dataset in which the features of the dataset are distributed across multiple organizations, but the label information is owned by a single organization. Here, the organization which has the label information is referred to as the guest client and those without label as host client [4]. A collaboration between general and specialized hospitals is an example of VFL. They may have the same patient's data, but the general hospital owns the patient's generic information (i.e., features), whereas the specialized hospital owns the patient's specific testing results (i.e., labels/ground truth). As a result, they can use VFL to jointly train a model that predicts a specific disease examined by the specialized hospital based on the general hospital's features [5]. For vertically partitioned data, privacy-preserving machine-learning algorithms have been proposed, such as the secure linear regression algorithm on vertically distributed data [6], privacy-preserving logistic regression [7] and verifiable privacy-preserving scheme (VPRF) based on vertical federated random forest [8]. Figure 1 [4] illustrates the basic protocol of the vertical federated learning approach, which involves the following major steps: (1) exchanging intermediate results, (2) computing gradients or loss, and (3) updating models on each of the clients. When naively following the protocol, every participating client has to communicate the intermediate results or updated gradients during every training iteration. The total communication for each client can easily grow significantly over the course of hundreds of thousands of training iterations

on large data sets. As a result, if communication bandwidth is limited or communication is expensive, FL can become ineffective [9].

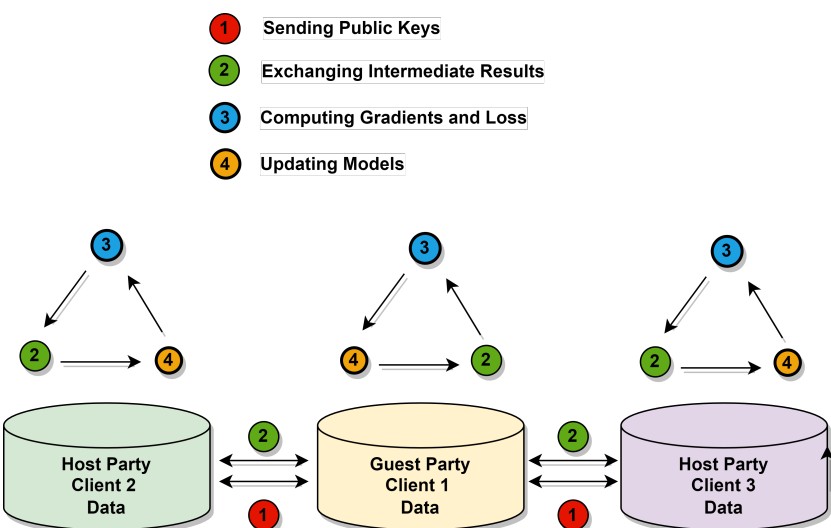

**Figure 1.** Iterative Federated Learning Environment (Adapted from [4]).

### 2.2. Communication-Efficient Federated Learning

In general distributed learning environments, communication costs have always been a constraint. Since federated learning is a form of distributed machine learning that deals with data privacy issues, this field has attracted a lot of interest in terms of improving communication efficiency. The studies that have been conducted related to communication-efficient FL have focused on the issue of huge communication rounds or bandwidth specifically in the HFL environment. The strategies that have been explored so far that could make communication in FL more efficient include choosing selective clients, reducing the number of model updates, and applying compression schemes to models. Client selection is a strategy for improving communication efficiency, while lowering costs by limiting the number of participants. As a result, just a portion of the parameters are updated over the communication round. Chen et al. [10] proposed a communication-efficient FL framework using a probabilistic device selection scheme such that only those clients are selected for model transmission that have higher probabilities to improve the convergence speed and training loss. Guha et al. [11] proposed a similar approach in which restrictions were imposed on the number of local models sent to the server for aggregation using various protocols (e.g., by random sampling, or by applying thresholds based on the amount of local data or local validation error).

As the number of communication rounds between the devices and the central server can be costly, reducing model updates is also a possible solution. Guha et al. [11] also suggested training local models on devices to completion instead of computing increments and then applying ensemble methods to effectively capture information regarding client-specific models, reducing communication rounds to one in the best case. Bui et al. [12] introduced a Partitioned Variational Inference (PVI) for probabilistic models, in which they train a Bayesian Neural Network (BNN) over an FL environment that allows for both synchronous and asynchronous model updates across many machines. Their proposed approach, combined with the integration of other methods, allows for more communication-efficient training of BNN on non-iid federated data. Some other studies that focus on reducing communication overhead in FL settings include a study by Li et al. [13], in which they designed a one-shot federated learning algorithm FedKT, which uses a knowledge transfer technique that outperforms the other state-of-the-art federated learning algorithms with a single communication round. Furthermore, Kasturi et al. [14] presented a federated fusion learning approach that allows distribution parameters of local data to be sent to the central server rather than model parameters. Synthetic data are generated at the central

server using those distribution parameters, which are then merged to train a global machine learning model. Hence, the communication between the client and server occurs in a single round.

The application of compression schemes to models is a widely used strategy to mitigate the communication cost problem in large-scale machine learning. Gradient quantization is a technique of quantizing the gradients into lower precision values to reduce the number of gradients transmitted. For instance, 1-bit SGD [15] and QSGD [16] can reduce the gradient to 10% of the uncompressed data. SignSGD [17] considers the case, wherein gradients are quantized using only $+1$ and $-1$, and shows its convergence by the aid of a majority vote of the clients. Some studies propose that applying quantization to the local or global model can help to improve communication efficiency in FL as well. Bui et al. [12] applied quantization to each local model before sending it to the server, reducing overall communication overhead. Similarly, quantization can be performed on the global model before broadcasting it to the clients for local updates. The Lossy FL algorithm (LFL) introduced in [18] quantizes the global model before it broadcasts and shares it across all the devices. Ref. [19] proposes an improved sign gradient descent [17] to replace conventional gradient descent in FL, which maintains the privacy of model parameters, while significantly decreasing the communication resource consumption. Gradient sparsification [20] is another compression technique that enforces transmitting $n$ out of $d$ elements at iteration $k$, where $d$ is the total number of elements in the gradient vector and $n$ is the number of the most important elements to send at iteration $k$. The work in [21] proposes the sending of only gradients larger than a pre-defined constant threshold. However, determining this threshold is a challenging task for gradient sparsification. The techniques Top-k AllReduce [22] and Deep Gradient Compression [20] were proposed to further improve compression efficiency. Gradient sparsification has also been proven to reduce the overall training time in FL [23] by adaptive selecting of the number of gradients or model parameters. Sattler et al. [9] proposed a sparse ternary compression (STC), an extension of the existing compression technique of top-k gradient sparsification [22] which was proven to outperform traditional federated averaging approach in case of bandwidth-constrained learning environments. However, experimental results demonstrating the effectiveness of most of these approaches in VFL setting were not observed.

Liu et al. [24] proposed a Federated Stochastic Block Coordinate Descent (FedBCD) algorithm for vertically partitioned data, in which each party conducts multiple local updates before each communication to effectively reduce the number of communication rounds among clients. Furthermore, Zhang et al. [25] developed an asynchronous stochastic quasi-Newton (AsySQN) framework for VFL, which performs descent steps scaled by approximate Hessian information, which converges much faster than Stochastic Gradient Descent (SGD)-based methods in practice, allowing for a significant reduction in the number of communication rounds. In [26], an asynchronous vertical federated learning framework with gradient prediction and double-end sparse compression is proposed, where the compression occurs at the local models to reduce training time as well as transmission cost. The existing compression techniques in FL are focused on gradient compression, which reduces training time and transmission costs, but still requires a sufficient number of communication rounds. On the other hand, in this paper, we propose a method based on the data compression technique that compresses local data of clients before aggregating it for final model training. Since no local data are exposed, privacy is protected, and the entire process is limited to a single communication round.

## 3. Proposed Method

The proposed method, as shown in Figure 2, is based on feature extraction techniques that reduce the dimensionality of data by removing redundancy. The feature extraction methods generally obtain new generated features by combining and transforming the original feature set, thus giving it a new latent representation. In a vertical federated setting, clients possessing relevant disjoint data are interested in training a global machine learning

model without exposing their raw data to each other. One of the clients is assumed to have labeled data (guest party) and the rest (host parties) have unlabeled data. The objective of the guest party is to be able to use the data from the host parties in order to perform better predictions for incoming new data, without compromising the privacy of data for itself as well as for other clients. Algorithm 1 shows this proposed method in pseudo code.

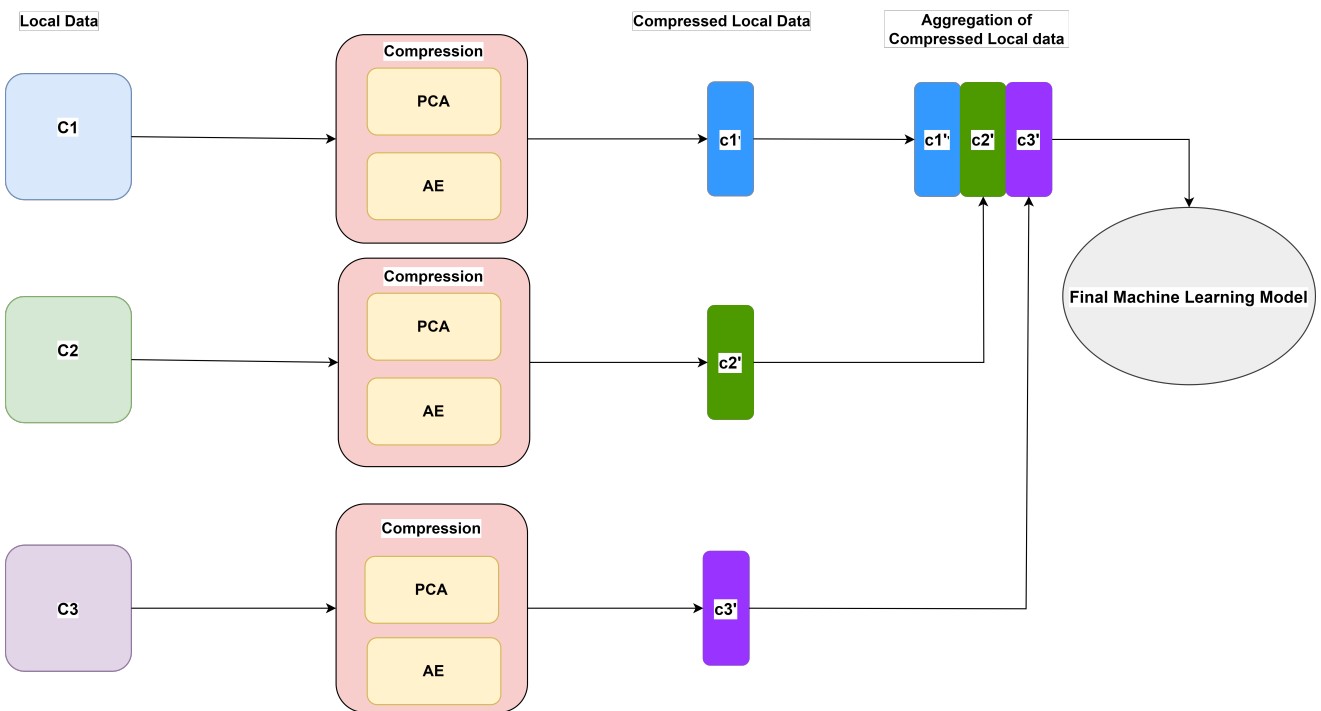

**Figure 2.** Architecture of Proposed Method.

---

**Algorithm 1** Communication-Efficient VFL

---

**Host Clients**
Local Data $X \leftarrow \{X_2...X_k\}$
**for** each client $i$ from 2..$k$ **do**
$\quad \hat{X}_i \leftarrow Latent(X_i)$
$\quad$ send $\hat{X}_i$ to Client 1
**Guest Client**
$\hat{X}_1 \leftarrow Latent(X_1)$
Receive $\hat{X} \leftarrow \{\hat{X}_2...\hat{X}_k\}$ $\qquad\qquad\qquad\qquad$ ▷ Latent representations from host clients
$D \leftarrow$ Merge $\hat{X}_1, \hat{X}$ $\qquad\qquad\qquad$ ▷ Aggregate latent representation of all clients
Train Model on $D$

---

### 3.1. First Step

The first step of the proposed method begins with performing feature extraction, also referred to as feature compression, on the local data of each client to generate latent representations of the local data. We used two feature extraction methods, i.e., Principal Component Analysis (PCA) and Autoencoders (AE), to perform the experiments for VFL and lastly, compared the performance of both techniques. The specifics of building the latent representations with each of these methods are described below.

#### 3.1.1. Principal Component Analysis

The goal of PCA is to explain the correlation structure of a set of features using a smaller set of linear combinations of these features [27]. These linear combinations are referred to as components. Suppose there are *m* features in a dataset, applying PCA to it

would yield *d* number of linear combinations in such a way that *d* components explain most of the information in the dataset. Hence, the dimension of the dataset is changed from *m* to *d* where $d < m$. The main steps for performing PCA on each of the client's local data are as follows:

- Standardizing the local data;
- Computing the covariance matrix of the features of the local data;
- Calculating eigenvalues and eigenvectors [28] for the covariance matrix;
- Sorting the eigenvectors by the magnitude of their corresponding eigenvalues;
- Determining *d*, the number of top principal components to select by methods such as imposing threshold on magnitude of eigenvalues or cumulative variance of the data.

PCA is performed on the local data of each client to simplify it and bring it into a new dimension while also retaining trends and patterns. Thus, latent or compressed representations of the local data are generated. PCA, as a dimension reduction technique, has several advantages. Computations are simple because PCA is based on linear algebra and the PCA components are simply linear combinations of the original features. Correlated features do not contribute significantly to the decision-making process in ML algorithms. As a result, removing them has no negative impact on the final output. PCA can efficiently eliminate these correlated features. However, PCA assumes that the principal components are a linear combination of the original features. If this does not hold, PCA will not yield sensible results. PCA is also sensitive to the scale of the features, so it is important that the data be standardized beforehand.

### 3.1.2. Autoencoders

Each client builds their own undercomplete autoencoder, which is used to extract important information from local data and convert it into latent data. These latent data are essentially the compressed knowledge representation of the original data. An undercomplete autoencoder is an unsupervised feed-forward neural network with the same inputs and outputs. The network is fully connected, with an encoder and a decoder. The encoder converts the input to a latent representation with lower dimensionality, which is then mapped by the decoder back to the original input. By minimizing the reconstruction loss, the network learns both the encoder and decoder weights during training. An autoencoder in Figure 3 has three major layers: input, hidden (latent representation), and output. Algorithm 2 describes a common method to train an autoencoder.

---

**Algorithm 2** Autoencoder Training

---

**procedure** $AE(e, b, X, \eta, \theta)$
  $X \leftarrow [X_1, X_2...X_n] \in R^{n*m}$ is the input matrix
  $e \leftarrow$ number of epochs
  $b \leftarrow$ number of batches
  $\eta \leftarrow$ learning rate
  $\theta \leftarrow \{W_h, W_o, b_h, b_o\}$
   where $[W_h, W_o] \in R^{n*d}, [b_h, b_o] \in R^d$
  **for** 0 to $e$ **do**
    **for** 0 to $b$ **do**
      $Z \leftarrow W_h{}^T X + b_h$
      $\hat{X} \leftarrow f(W_o^T Z + b_o)$
      $L(X, \hat{X}) \leftarrow ||\hat{X} - X||^2$
      $g \leftarrow$ compute gradients of $L(X, \hat{X})$ with respect
      to $\theta$
      **for** $\theta_i, g_i$ in $(\theta, g)$ **do**
        $\theta_i \leftarrow \theta_i - \eta * g_i$

---

Autoencoders are able to deal with linear as well as non-linear data depending on the choice of the activation function, unlike PCA. However, autoencoders train through

gradient descent and are slower comparatively. Autoencoders with a single layer and linear activation function perform the same as PCA. In the case of autoencoders with multiple layers and non-linear activation functions, they need to be carefully designed and controlled by regularisation to avoid overfitting.

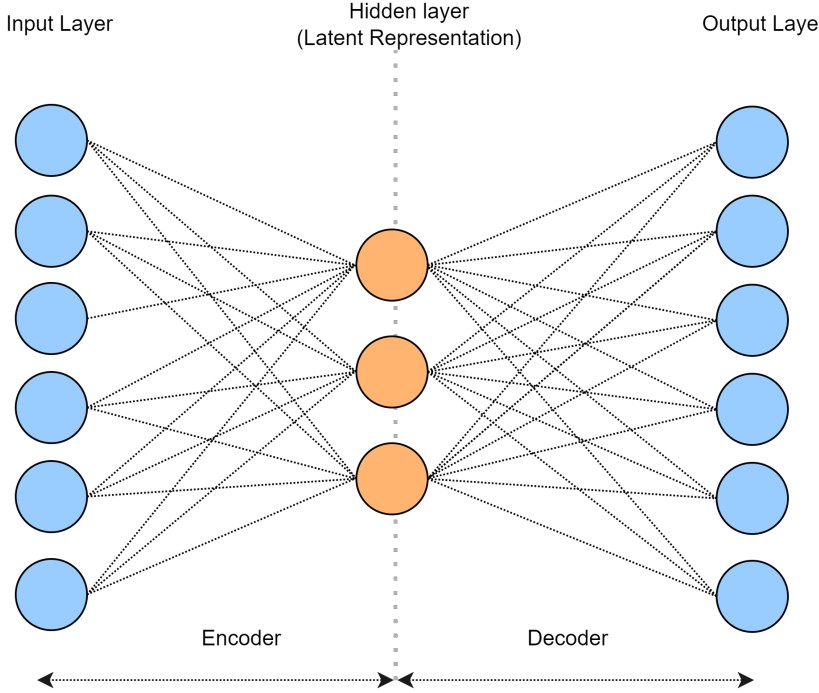

**Figure 3.** Undercomplete Autoencoder.

### 3.2. Second Step

Once all the clients have generated the latent representations of their local data, in the next step, these latent data from all the clients are aggregated to train the global model. In typical federated settings, the aggregation of local models is performed by a central server, but in this case, the data aggregation is performed by the guest client that has the target variable. The aggregated latent data are subsequently used to train the final global model which makes predictions for new data in the future. Each time the guest party needs to perform a prediction on new data, it requests from the other clients their latent data. The guest party receives the latent data, aggregates them and can perform the task using the final global model.

The latent data differ significantly from the original data of the clients but still contain relevant important information. As a result, sharing latent data from clients poses no risk of raw data exposure, while also improving performance.

## 4. Experimental Setup

This section describes the experimental setup used to evaluate the proposed method. For performing the evaluation, the classification problem has been considered for simplicity.

### 4.1. Datasets

We tested our proposed method on several datasets in Table 1 from the UCI repository, as well as other publicly available sources. One of the criteria for selecting the datasets was that they be publicly available. Second, the size of the datasets was taken into account. We chose the datasets in such a way that the number of samples as well as the features ranged from small to large, so that the robustness of the proposed method could be evaluated.

- **Adult Income Dataset:** The adult income dataset [29] includes two labels: whether or not a person earns more than $50,000 per year.

- **Heart Disease Dataset:** The dataset [30] contains medical information of patients across different hospitals. Predictions are made between patients with or without heart disease.
- **Wine Quality Dataset:** The dataset [31] contains chemical properties of red and white variants of the Portuguese "Vinho Verde" wine. Classification are made between red and white wine.
- **Rice MSC Dataset:** The dataset [32] contains morphological and shape features of five types of rice varieties. Classification of the rice variety is to be performed.

**Table 1.** Overview of Datasets.

| Dataset | Instances | Features |
|---|---|---|
| Adult Income Dataset | 22,404 | 14 |
| Heart Disease Prediction Dataset | 302 | 13 |
| Wine Quality Dataset | 5329 | 12 |
| Rice MSC Dataset | 75,000 | 106 |

All the datasets were pre-processed accordingly taking into account factors such as missing values, duplication, and class imbalance. Over-sampling was used to balance datasets with imbalanced classes. Moreover, One-Hot-Encoding was performed on the categorical features of the datasets (e.g., *Adult Income Dataset*). To simulate a vertically federated environment, the datasets were vertically divided into nearly equal three partitions, each representing data for 3 clients. It is assumed that one of the clients (guest party) has the labels or ground truth. We used the following datasets to perform the evaluation.

### 4.2. Fitting and Selecting Components in PCA

The local data of each of the clients is split into train (70%) and test set (30%). For the implementation of PCA the *sklearn.decomposition.PCA* class has been used. The *fit()* function is applied on the train set at each client end to compute the eigen vectors. PCA components are selected in such a way that 90% cumulative data variance is captured. Later, the *transform()* function is used on both the train and test set to apply the projections of the eigen vectors obtained. The *transform()* function on the train and test reduces the dimension of the original data while also explaining 90% variance.

### 4.3. Training Autoencoder

Local data of each of the clients is split into 3 sets; training set (70%), validation set (15%), and testing set (15%). The train, validation, and test splits are performed to prevent the autoencoder model from overfitting and to accurately evaluate it. The hyperparameters used for the training of autoencoders were:

- **Encoding dimension:** This refers to the number of nodes at the bottleneck and is the dimensionality of the encoded representation of the data. For our experiments, a reduction percentage (25%, 50%, 75%) was provided during training.
- **Learning rate:** The learning rate was varied between 0.1 to 0.001.
- **Batch size:** The batch size is the number of samples per gradient update. The following values were tested: 10, 100, and 1000.
- **Epoch:** Experiments were tested with 1000 epochs
- **Optimizer:** This is the optimization algorithm that is used to minimize the loss. The "Adam" optimizer was used for the experiments.
- **Loss Function:** This is the error function which is evaluated during the neural network training stage and the training strives to minimize the losses. The "mean_squared_error" loss function was used.

*4.4. Aggregated Model Training*

After latent representations are generated from all the clients, they are aggregated by the guest client (C1) for final model training. The aggregated data are then divided into three groups: training (70%), validation (15%), and testing (15%). The *Adult Income*, *Heart Disease*, and *Wine Quality* datasets were used for a binary classification task. The *Logistic Regression* model was used to train the final model with these datasets. On the other hand, the *Rice MSC dataset* was used for a multiclass classification task, and the *Support Vector Classifier (SVC)* model was used to train the final model with this dataset.

## 5. Results

In this section, the experimental results obtained from the proposed method are presented. The performance metrics (Accuracy and F1-Score) obtained through the proposed method have been compared against the centralized learning system and also the guest party's learning system using only its local data.

*5.1. Performance using PCA*

Tables 2–5 provide a comparison of the proposed method's performance with centralized training and guest client training with its local data. Furthermore, they demonstrate how the proposed method's performance is affected when PCA is not performed on the local data of guest client C1.

**Table 2.** Experimental Results on Adult Income Dataset.

|  | Accuracy | F1 Score |  |
|---|---|---|---|
| Centralized Training with unpartitioned data | 81.45 | 81.46 |  |
| Guest party with local data | 68.88 | 68.87 |  |
| Performance after PCA at all clients | 78.46 | 78.44 |  |
| Performance after PCA at all clients except C1 | 62.22 | 62.19 |  |
| **Overview of Feature Dimensions** | | | |
|  | **Client 1** | **Client 2** | **Client 3** |
| Feature dimension before PCA | 26 | 34 | 43 |
| Feature dimension after PCA | 20 | 24 | 38 |

**Table 3.** Experimental Results on Heart Disease Prediction Dataset.

|  | Accuracy | F1 Score |  |
|---|---|---|---|
| Centralized Training with unpartitioned data | 86.81 | 86.21 |  |
| Guest party with local data | 81.31 | 81.10 |  |
| Performance after PCA at all clients | 81.49 | 81.51 |  |
| Performance after PCA at all clients except C1 | 81.24 | 81.23 |  |
| **Overview of Feature Dimensions** | | | |
|  | **Client 1** | **Client 2** | **Client 3** |
| Feature dimension before PCA | 4 | 5 | 4 |
| Feature dimension after PCA | 2 | 3 | 3 |

**Table 4.** Experimental Results on Wine Quality Dataset.

|  | Accuracy | F1 Score |
|---|---|---|
| Centralized Training with unpartitioned data | 75.10 | 74.29 |
| Guest party with local data | 66.10 | 58.95 |
| Performance after PCA at all clients | 74.79 | 72.18 |
| Performance after PCA at all clients except C1 | 74.79 | 72.08 |
| **Overview of Feature Dimensions** | | |
|  | **Client 1** | **Client 2** | **Client 3** |
| Feature dimension before PCA | 4 | 4 | 4 |
| Feature dimension after PCA | 3 | 2 | 2 |

**Table 5.** Experimental Results on Rice MSC Dataset.

|  | Accuracy | F1 Score |
|---|---|---|
| Centralized Training with unpartitioned data | 88.20 | 88.17 |
| Guest party with local data | 73.83 | 73.40 |
| Performance after PCA at all clients | 84.92 | 84.91 |
| Performance after PCA at all clients except C1 | 79.92 | 79.53 |
| **Overview of Feature Dimensions** | | |
|  | **Client 1** | **Client 2** | **Client 3** |
| Feature dimension before PCA | 26 | 42 | 38 |
| Feature dimension after PCA | 7 | 7 | 9 |

The experimental results using the proposed method with PCA show that the aggregated model outperforms the local model of the guest client across all datasets. However, if PCA is not performed on the local data for the guest client, performance varies across datasets. For example, the accuracy and F1 score improve for the *Wine Quality Dataset* (Table 4), but decrease for the other datasets. This might be a result of the reduction of noise in the dataset due to the application of PCA on it.

*5.2. Performance using Autoencoders*

Tables 6–9 provide a similar comparison to that shown in Section 5.1, but with an undercomplete autoencoder as a feature extraction technique. The table also provides an overview of how well the autoencoders perform when encoding local data of the clients. To gain a better understanding and observe the effect on overall performance, experiments were also carried out by varying the compression rates/encoding dimensions (25, 50, and 75 percent) of the autoencoders. A common observation across all datasets is that the performance of the aggregated model after applying the proposed method degrades as the compression rate of the autoencoder is reduced.

**Table 6.** Experimental Results on Adult Income Dataset.

|  | Accuracy (%) | | | | | F1 Score (%) |
|---|---|---|---|---|---|---|
| Centralized Learning | 81.45 | | | | | 81.46 |
| Guest Party Performance | 68.88 | | | | | 68.87 |
| | **Autoencoder Performance** | | | | | |
| | **Compression 25%** | | **Compression 50%** | | **Compression 75%** | |
| | Training Loss | Validation Loss | Training Loss | Validation Loss | Training Loss | Validation Loss |
| C1 | 74.27 | 75.69 | 76.72 | 77.15 | 80.14 | 80.66 |
| C2 | 0.0189 | 0.0194 | 0.0326 | 0.0340 | 0.0698 | 0.0699 |
| C3 | 9.84 | 10.06 | 36.14 | 37.22 | 45.15 | 46.61 |
| | **Performance on Aggregated Latent Data** | | | | | |
| | **Compression 25%** | | **Compression 50%** | | **Compression 75%** | |
| | Accuracy (%) | F1 Score (%) | Accuracy (%) | F1 Score (%) | Accuracy (%) | F1 Score (%) |
| Encoding C1 | 77.88 | 77.89 | 75.19 | 75.20 | 75.72 | 75.73 |
| Without Encoding C1 | 80.83 | 80.84 | 79.79 | 79.79 | 79.31 | 79.32 |

**Table 7.** Experimental Results on Heart Disease Prediction Dataset.

| | Accuracy (%) | | | | F1 Score (%) | |
|---|---|---|---|---|---|---|
| Centralized Learning | 86.81 | | | | 86.21 | |
| Guest Party Performance | 81.32 | | | | 81.10 | |

| | Autoencoder Performance | | | | | |
|---|---|---|---|---|---|---|
| | Compression 25% | | Compression 50% | | Compression 75% | |
| | Training Loss | Validation Loss | Training Loss | Validation Loss | Training Loss | Validation Loss |
| C1 | 15.78 | 15.46 | 23.60 | 22.12 | 33.05 | 32.15 |
| C2 | 7.98 | 7.98 | 16.13 | 15.16 | 19.98 | 21.12 |
| C3 | 5.87 | 5.46 | 8.87 | 7.72 | 10.25 | 9.98 |

| | Performance on Aggregated Latent Data | | | | | |
|---|---|---|---|---|---|---|
| | Compression 25% | | Compression 50% | | Compression 75% | |
| | Accuracy (%) | F1 Score (%) | Accuracy (%) | F1 Score (%) | Accuracy (%) | F1 Score (%) |
| Encoding C1 | 78.76 | 78.77 | 78.17 | 78.19 | 75.29 | 75.31 |
| Without Encoding C1 | 81.84 | 81.84 | 80.98 | 80.99 | 80.74 | 80.75 |

**Table 8.** Experimental Results on Wine Quality Dataset.

| | Accuracy (%) | | | | F1 Score (%) | |
|---|---|---|---|---|---|---|
| Centralized Learning | 75.10 | | | | 74.29 | |
| Guest Party Performance | 66.10 | | | | 58.95 | |

| | Autoencoder Performance | | | | | |
|---|---|---|---|---|---|---|
| | Compression 25% | | Compression 50% | | Compression 75% | |
| | Training Loss | Validation Loss | Training Loss | Validation Loss | Training Loss | Validation Loss |
| C1 | 0.2049 | 0.2709 | 0.2157 | 0.2387 | 0.3624 | 0.3819 |
| C2 | 0.5218 | 0.5338 | 0.6073 | 0.6169 | 0.8411 | 0.8732 |
| C3 | 0.2902 | 0.3007 | 0.3034 | 0.3407 | 0.5148 | 0.5332 |

| | Performance on Aggregated Latent Data | | | | | |
|---|---|---|---|---|---|---|
| | Compression 25% | | Compression 50% | | Compression 75% | |
| | Accuracy (%) | F1 Score (%) | Accuracy (%) | F1 Score (%) | Accuracy (%) | F1 Score (%) |
| Encoding C1 | 78.43 | 78.45 | 78.10 | 78.15 | 75.36 | 75.39 |
| Without Encoding C1 | 81.69 | 81.70 | 81.29 | 81.30 | 81.24 | 81.25 |

**Table 9.** Experimental Results on Rice MSC Dataset.

| | Accuracy (%) | | | | F1 Score (%) | |
|---|---|---|---|---|---|---|
| Centralized Learning | 88.20 | | | | 88.17 | |
| Guest Party Performance | 73.83 | | | | 73.40 | |

| | Autoencoder Performance | | | | | |
|---|---|---|---|---|---|---|
| | Compression 25% | | Compression 50% | | Compression 75% | |
| | Training Loss | Validation Loss | Training Loss | Validation Loss | Training Loss | Validation Loss |
| C1 | 61.46 | 62.98 | 77.52 | 79.43 | 93.41 | 96.24 |
| C2 | 42.39 | 43.73 | 61.48 | 63.68 | 78.66 | 79.91 |
| C3 | 18.21 | 19.18 | 21.57 | 22.05 | 31.06 | 34.37 |

| | Performance on Aggregated Latent Data | | | | | |
|---|---|---|---|---|---|---|
| | Compression 25% | | Compression 50% | | Compression 75% | |
| | Accuracy (%) | F1 Score (%) | Accuracy (%) | F1 Score (%) | Accuracy (%) | F1 Score (%) |
| Encoding C1 | 78.57 | 78.80 | 71.78 | 71.42 | 71.60 | 70.89 |
| Without Encoding C1 | 74.23 | 73.90 | 74.17 | 73.85 | 74.02 | 73.79 |

In the case of the *Adult Income Dataset* in Table 6, it is seen that the autoencoder performs well during the compression of local data of other clients compared to the guest client. As the autoencoder becomes quite lossy when encoding the local data of the guest client, it stands to reason that the aggregated model's performance would be improved if the local data of the guest client were not encoded with the autoencoder.

A similar trend is noticed in the case of the *Heart Disease Dataset* as well (Table 7). However, it is only in this case that the autoencoders' training loss is greater than their corresponding testing losses, indicating overfitting. As autoencoders require a large number of samples to be trained, a small number of samples tends to overfit it. Overfitting is possible due to the small number of samples in the *Heart Disease Dataset*.

The losses of the autoencoders across all clients are very low in the case of the *Wine Quality Dataset*, indicating that the autoencoders can compress the raw local data with max-

imum information even at the highest compression rate (75%). As a result, the aggregated model constructed using the proposed approach outperforms both the local model of the guest client and even the central model.

An important observation is that when the dimension of the dataset is too large, as in the case of the *Rice MSC Dataset* (Table 5), the autoencoders become lossy. The high losses of the autoencoders indicate that the local data are not being compressed, having most of the information. Hence, a drop in the performance of the aggregated model is observed. However, it is clear that if the autoencoder losses are minimized, the proposed method will perform well, just as it did in the other datasets discussed earlier. The performance of the autoencoders can be improved by increasing the number of hidden layers and properly tuning the hyperparameters while training.

## 6. Conclusions and Future Work

In this paper, we propose a communication-efficient approach for vertical federated learning in which clients are assumed to have different features but are interested in collaborating to build a global ML model. Our approach mimics the concept of centralized learning (aggregating local data from clients on a single site), but also ensures privacy by compressing the data. The experimental results show that our proposed method is sufficiently robust to be applied to data sets of varying sizes. As all algorithms used in our proposed method have been proven to converge [33–36], the proposed method will converge as well. Furthermore, it has been demonstrated that the performance of the final model using our proposed method outperforms the local model of the guest client, which was the goal of the study. However, depending on the type of data, a proper decision on the compression technique to be used must be made. There is room for improvement by incorporating feature selection methods on the aggregated compressed data at the client end so that the guest client only needs the important features from the host client when making predictions on new data in the future. In our future work, a detailed comparison between the performance of our proposed approach and the existing leading ones related to communication-efficient vertical federated systems will be included.

**Author Contributions:** Conceptualization, A.K., M.t.T. and A.W.; Data curation, A.K.; Investigation, A.K.; Methodology, A.K., M.t.T. and A.W.; Supervision, M.t.T. and A.W.; Writing—original draft, A.K.; Writing—review and editing, M.t.T. and A.W. All authors have read and agreed to the published version of the manuscript.

**Funding:** This research received no external funding.

**Institutional Review Board Statement:** Not applicable.

**Informed Consent Statement:** Not applicable.

**Data Availability Statement:** The datasets used in this study are publicly available in the repositories at https://archive.ics.uci.edu/ml (Date accessed: 19 June 2022) and https://www.muratkoklu.com/datasets/ (Date accessed: 19 June 2022).

**Conflicts of Interest:** The authors declare no conflict of interest.

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
