# Peer review of "Communication-Efficient Vertical Federated Learning"

_algorithms, doi:10.3390/a15080273_

Round 1
Reviewer 1 Report
The paper proposes a communication-efficient approach that compresses the local data of clients. In this way, the privacy of these data is preserved while it is compressed. The experiments show the effectiveness of the proposed algorithm.
However, the paper has some issues to be solved.
First, the paper lacks proofreading, making the paper confusable. For example, double ‘without’ are in the abstract. Besides, do not put all figures/tables on one page. The tables should appear when they are mentioned. Several other issues need to be checked in the paper.
What’s more, the content in the related work is not satisfactory. Vertical federated learning belongs in the category of federated learning; thus, there is no need to introduce the related work of federated learning. The paper should discuss other methods of communication-efficient systems, not limited to federated learning.
Even so, I would like to suggest the decision of accepting with revision due to the extensive experiments and clear presentations of the methods.
Author Response
Thank you for giving us the opportunity to submit a revised draft of the manuscript “Communication-Efficient Vertical Federated Learning” for publication in the Algorithms Journal. We appreciate the time dedicated by you to providing feedback on our manuscript and are grateful for the insightful comments and valuable improvements to our paper. We have incorporated the suggestions provided. Those changes are highlighted within the manuscript. Please see below for a point-by-point response to the comments and concerns.
Comment 1: The paper proposes a communication-efficient approach that compresses the local data of clients. In this way, the privacy of this data is preserved while it is compressed. The experiments show the effectiveness of the proposed algorithm.
Author’s Response: Thank you.
Comment 2: The paper lacks proofreading, making the paper confusable. For example, double ‘without’ are in the abstract. Besides, do not put all figures/tables on one page. The tables should appear when they are mentioned. Several other issues need to be checked in the paper.
Author’s Response: Thank you for pointing this out. We have gone through the manuscript thoroughly and made sure to fix any wording issues, typos, and grammatical errors in the revised manuscript. We have also better positioned the figures and tables.
Comment 3: The content in the related work is not satisfactory. Vertical federated learning belongs in the category of federated learning; thus, there is no need to introduce the related work of federated learning. The paper should discuss other methods of communication-efficient systems, not limited to federated learning
Author’s Response: Thank you for your valuable suggestion. We have revised the related works section by discussing existing communication-efficient systems, which are mostly based on model compression techniques.
Reviewer 2 Report
In this paper, the authors proposed a model compression method to improve the communication efficiency of vertical federated learning. The overall performance is acceptable. Generally speaking, the paper is easy to follow and logically clear.
However, there are some issues.
1) The contributions and novelties should be better refined to make the paper stand out.
2) It would be better if the authors can theoretically prove the convergence of the model.
3) It seems the authors did not compare the proposed model with existing leading ones.
4) The presentation can be improved, for example, user three-line table to better present the results.
5) The authors may discuss other model compression models as related work or comparison, for instance, SignSGD (FL-SEC: Privacy-Preserving Decentralized Federated Learning Using SignSGD for the Internet of Artificially Intelligent Things). In addition, the authors may refer to a Survey with more detailed discussion on compression in FL (Asynchronous federated learning on heterogeneous devices: A survey).
6) The authors may consider polishing the paper from the perspectives of wording issues, typos, and grammar errors.
Author Response
Thank you for giving us the opportunity to submit a revised draft of the manuscript “Communication-Efficient Vertical Federated Learning” for publication in the Algorithms Journal. We appreciate the time dedicated by you to providing feedback on our manuscript and are grateful for the insightful comments and valuable improvements to our paper. We have incorporated most of the suggestions provided. Those changes are highlighted within the manuscript. Please see below for a point-by-point response to the comments and concerns.
Comment 1: The authors proposed a model compression method to improve the communication efficiency of vertical federated learning. The overall performance is acceptable. Generally speaking, the paper is easy to follow and logically clear.
Author’s Response: Thank you.
Comment 2: The contributions and novelties should be better refined to make the paper stand out.
Author’s Response: Thank you for the suggestion. We have mentioned our contributions in the revised manuscript clearly in a way so that it is easy to understand.
Comment 3: It would be better if the authors can theoretically prove the convergence of the model.
Author’s Response: Thank you for the comment. We would like to mention that our proposed approach uses existing feature extraction techniques (PCA and Autoencoders) which compresses the local data of participating clients. The aggregated compressed local data is used to train an existing machine-learning model (In our case Logistic Regression and Support Vector Classifier). For both the feature extraction techniques and the machine learning model, the proof of convergence is well established. Hence, we feel it is not necessary to have separate proof for the convergence of our proposed approach.
Comment 4: It seems the authors did not compare the proposed model with existing leading ones
Author’s Response: Thank you for your comment. We agree that it would be a good idea to compare the proposed approach with the existing leading ones. But, it will take a considerable amount of time to run the experiments and discuss the observations. Hence, we plan to perform a detailed comparison in future work.
Comment 5: The presentation can be improved, for example, user three-line table to better present the results.
Author’s Response: The table presentation has been improved. Since our tables include subheadings, the three-line table format is not suitable. But, we have used a similar format.
Comment 6: The authors may discuss other model compression models as related work or comparison, for instance, SignSGD (FL-SEC: Privacy-Preserving Decentralized Federated Learning Using SignSGD for the Internet of Artificially Intelligent Things). In addition, the authors may refer to a Survey with more detailed discussion on compression in FL (Asynchronous federated learning on heterogeneous devices: A survey).
Author’s Response: Thank you for the detailed suggestion. We have revised the related works section by discussing existing works on model compression.
Comment 7: The authors may consider polishing the paper from the perspectives of wording issues, typos, and grammar errors.
Author’s Response: We have gone through the manuscript thoroughly and made sure to fix any wording issues, typos, and grammatical errors in the revised manuscript.
Reviewer 3 Report
The quality and contents of this manuscript look okay. I feel that some experts in the specific research domain can judge better its contributions.
Author Response
Thank you for your time to review our manuscript.
No comments to address.
Round 2
Reviewer 2 Report
After reading through the revised manuscript, the reviewer believed that the author has addressed all my concerns. It would be the best if the authors can further polish the paper with regards to typos and wording issues. Besides, the authors are suggested to discuss "Asynchronous federated learning on heterogeneous devices: A survey" if possible. In summary, the paper is suitable for acceptance for publication.